# Template-Free Preparation of α-Ni(OH)_2_ Nanosphere as High-Performance Electrode Material for Advanced Supercapacitor

**DOI:** 10.3390/nano12132216

**Published:** 2022-06-28

**Authors:** Rongrong Zhang, Qian Tu, Xianran Li, Xinyu Sun, Xinghai Liu, Liangzhe Chen

**Affiliations:** 1School of Electronic Information Engineering, Jingchu University of Technology, Jingmen 448000, China; 200805001@jcut.edu.cn (R.Z.); tq010406@163.com (Q.T.); ggll925597@163.com (X.L.); 2Research Center of Graphic Communication, Printing and Packaging, Wuhan University, Wuhan 430079, China

**Keywords:** α-Ni(OH)_2_, nanosphere, supercapacitor, template-free, morphology engineering

## Abstract

Although it is one of the promising candidates for pseudocapacitance materials, Ni(OH)_2_ is confronted with poor specific capacitance and inferior cycling stability. The design and construction of three-dimensional (3D) nanosphere structures turns out to be a valid strategy to combat these disadvantages and has attracted tremendous attention. In this paper, a 3D α-Ni(OH)_2_ nanosphere is prepared via a facile and template-free dynamic refluxing approach. Significantly, the α-Ni(OH)_2_ nanosphere possesses a high specific surface area (119.4 m^2^/g) and an abundant porous structure. In addition, the as-obtained α-Ni(OH)_2_ electrodes are investigated by electrochemical measurements, which exhibit a high specific capacitance of 1243 F/g at 1 A/g in 6 M KOH electrolyte and an acceptable capacitive retention of 40.0% after 1500 charge/discharge cycles at 10 A/g, which can be attributed to the sphere’s unique nanostructure. Furthermore, the as-assembled Ni(OH)_2_-36//AC asymmetric supercapacitor (ASC) yields a remarkable energy density of 26.50 Wh/kg, with a power density of 0.82 kW/kg. Notably, two ASCs in series can light a 2.5 V red lamp sustainably for more than 60 min, as well as power an LED band with a rated power of 25 W. Hence, this 3D α-Ni(OH)_2_ nanosphere may raise great potential applications for next-generation energy storage devices.

## 1. Introduction

The supercapacitor (SC), also known as an electrochemical capacitor, is deemed to be one of the most promising energy storage devices to address ever-increasing environmental issues and the energy crisis because of its the fast charge and discharge ability, high power density and long durability [1,2,3,4,5]. On the ground level of energy storage mechanisms, SCs are grouped into two categories: pseudocapacitance and electric double-layer capacitance (EDLC) [6,7]. As a common EDLC material, the activated carbon (AC) has tremendous applications owing to its low prices and abundant sources [8,9]. However, AC-based SCs suffer from poor specific capacitance and lower operating voltages (usually less than 1.0 V) [10,11]. Of the various strategies, the most effective candidate to remove these limitations is the fabrication of an asymmetric system, namely, the activated carbon and pseudocapacitance materials serve as negative and positive electrodes, respectively, which enables utilization of the advantages of two types of SCs. Hence, the primary task ahead is to focus on the study of positive electrode materials with excellent electrochemical performances [12,13]. 

In recent years, transition-metal hydroxide, especially Ni(OH)_2_, has attracted intensive concerns as a result of its low cost, remarkable energy density and theoretical specific capacitance [14,15,16]. Even so, the shortcomings of inferior electric conductivity and poor rate capability are becoming serious obstacles to the widespread application of supercapacitors [17,18,19,20,21]. Hence, more explorations are essential for coping with these limitations. Designing and constructing nanostructured Ni(OH)_2_ materials turns out to be one of the most attractive strategies. Some researchers have pointed out that the excellent electrochemical properties of Ni(OH)_2_-based electrodes rely on nanostructure morphology, which is associated with the preparation strategy. For instance, Sun et al. [22] synthesized few-layered Ni(OH)_2_ nanosheets (4–5 nm in thickness), which show high specific capacitance and rate capacity in both electrode and supercapacitor. Niu et al. [23] prepared a unique ZnO nanofiber@Ni(OH)_2_ core–shell heterostructure with a significant specific capacitance (of 2218 F/g at 2 mV/s). On the other hand, the nanostructure Ni(OH)_2_ can create a larger specific surface area and supply more active sites, generating a shorter ion transfer pathway and higher capacitance [24]. In addition, since the topic of carbon neutrality has become prominent, designing and synthesizing a nanostructured Ni(OH)_2_ in a simple, convenient and low-cost way is very meaningful.

More recently, Yang et al. [25] prepared a 3D MXene-based microsphere via the spray pyrolysis and selenization process, which exhibited an enhanced capacity and capacity retention. Inspired by this study, we design and propose a facile approach to fabricating a nanostructured α-Ni(OH)_2_ nanosphere via a modified co-precipitation method without any template reagent, i.e., dynamic refluxing route. Compared with traditional methods, this present route is superior because of the simplified process (only one step) and freedom from high pressure (at normal pressure). It should be emphasized that the surface morphology and architecture can be easily controlled by adjusting the reaction times. Importantly, this 3D-nanostructured α-Ni(OH)_2_ nanosphere displays a distinguished faradaic electrochemical performance with high specific capacitance and receptible cycling retention, which can be attributed to its large specific surface area and abundant mesoporous structure provided by the nanoball construction. Moreover, the assembled asymmetric supercapacitor (Ni(OH)_2_-36//AC ASC) delivers remarkable power density and energy density, which is competitive with the Ni(OH)_2_-based system in previous reports. Surprisingly, two ASCs in the series show good practical performances, which can not only light up a red lamp with a voltage of 2.5 V for more than 60 min, but also power an LED band with a rated power of 25 W. Hence, this paper proposes a facile approach for preparing the advanced electrode material for next-generation energy storage devices.

## 2. Materials and Methods

### 2.1. Materials and Chemicals

Nickel chloride (NiCl_2_·6H_2_O), urea, acetone, ethanol, potassium hydroxide (KOH) and hydrochloric acid (HCl) were purchased from Shanghai Macklin Biochemical Co., Ltd. (Shanghai, China). All chemical reagents were of analytical purity and used without further processing. Nickel foam (1 mm in thickness) and polytetrafluoroethylene (PTFE) were received from Cyber Electro Chemical Materials. Acetylene black was obtained from Yilongsheng Energy Technology Co., Ltd. (Suzhou, China). and activated carbon (AC) (5 ± 1 μm in particle size, 1800 ± 100 m^2^/g in specific surface area, 1.0–1.2 cm^3^/g in pore volume and 0.38–0.40 g/cm^3^ in tap density) was bought from Nanjing/Jiangsu XFNANO Materials Tech Co., Ltd. (Nanjing, China). The deionized water was used throughout the experiments. 

### 2.2. Synthesis of the 3D α-Ni(OH)_2_ Nanosphere

The α-Ni(OH)_2_ nanosphere is synthesized via a facile dynamic reflux method. Briefly, 1.26 mmol of NiCl_2_·H_2_O and 12.6 mmol of urea were dissolved in 50 mL of deionized water to obtain a homogeneous solution and were placed in a three-necked flask, which was heated at 105 °C for 24~60 h. Then, the filter cake was washed with deionized water until neutral and dried in a vacuum for another 24 h. According to the reaction time, the obtained Ni(OH)_2_ compositions were labeled as Ni(OH)_2_-24, Ni(OH)_2_-36, Ni(OH)_2_-48 and Ni(OH)_2_-60, respectively. 

### 2.3. Preparation of the α-Ni(OH)_2_ Electrodes

The electrodes, including the α-Ni(OH)_2_ electrode (anode) and AC electrode (cathode), were prepared by a screen-printing route, as shown in Appendix A. For screen-printed electrodes, an 80 wt% α-Ni(OH)_2_ composite or AC, 10 wt% acetylene black and 10 wt% PTFE/ethanol emulsion were mixed by stirring continuously to form a homogeneous ink. Then, the resulting inks were forced through the screen plate (100 meshes) and printed on nickel foam with a square pattern of 1 cm × 1 cm, followed by drying at 80 °C for 3 h. Finally, the product was treated with pressure under 8 MPa for 1 min. Importantly, the loading mass of the electrode can be regulated by adjusting the printing times or ink concentration, and the loading mass of the working electrode was ~3.0 mg in this work.

### 2.4. Fabrication of the Asymmetrical Supercapacitor

The asymmetrical supercapacitor (ASC) was assembled on a Swagelok container with a “sandwich” structure. In general, the positive electrode, negative electrode and cellulose membrane were dipped into a 6 mol/L KOH solution for 5 min. Then, two electrodes were assembled face-to-face with a cellulose membrane between them. Finally, this “sandwich” was clamped by two stainless steel sheets under gentle pressure. 

In order to obtain the best electrochemical performances, the mass ratios of Ni(OH)_2_ and AC were designed via the charge–balance (*q*^+^
*= q*^−^) of the electrode materials according to equations [26,27]:(1)$$q=C_{m}\times \Delta V\times m$$
(2)$$\frac{m^{+}}{m^{-}}=\frac{C^{-}\times \Delta V^{-}}{C^{+}\times \Delta V^{+}}\;$$
where *C_m_*, ∆*V* and *m* represent the specific capacitance of the positive and negative electrodes (F/g), potential windows (V) and the mass of active materials on positive and negative electrodes (g), respectively.

### 2.5. Fabrication of the Asymmetrical Supercapacitor

For the single electrode, the electrochemical performances were tested in a three-electrode cell with a 6 mol/L KOH electrolyte. The Ni(OH)_2_ or AC electrode, Hg/HgO electrode and platinum plate served as the working electrode, reference electrode and counter electrode, respectively. For the asymmetrical supercapacitor (ASC), the electrochemical test was implemented in a two-electrode system, namely, the Ni(OH)_2_ electrode and AC electrode served as the positive electrode and negative electrode, respectively. Then, cyclic voltammetry (CV), galvanostatic charge–discharge (GCD) and electrochemical impedance spectroscopy (EIS) and cycling stability test were measured at 25 °C. The energy density (*E*) and power density (*P*) of the ASC were calculated based on the following equations [28,29]:(3)$$E=\frac{1}{2}C_{m}\Delta V^{2}$$
(4)$$P=\frac{E}{t}\;$$
where *C_m_* is the specific capacitance of the ASC based on the GCD curves (F/g), ∆*V* is the voltage window (V), and *t* is the discharge time (h). 

### 2.6. Characterization

The phase structure of the samples was analyzed by an X-ray diffractometer (XRD, RIGAKU Miniflex600, Tokyo, Japan) with Cu Kα radiation (λ = 1.5406 Å) from 5 to 80°. The morphology was characterized by a scanning electron microscope (SEM, Zeiss SIGMA 500, Oberkochen, Germany) and a transmission electron microscope (TEM, FEI Tecnai G2 F20, Hillsboro, OR, USA) operated at 200 kV. The surface valence bond and functional group information were identified by X-ray photoelectron spectroscopy (XPS, ThermoFisher EscaLab250Xi, Waltham, MA, USA) with a monochromatic Al Kα radiation (1486.6 eV) and Fourier transformed infrared spectrometer (FT-IR, ThermoFisher Nicolet 5700, Waltham, MA, USA) in the range of 500–4000 cm^−1^. The specific surface area of the sample was studied by the Brunauer–Emmett–Teller method (BET, Micromeritics ASAP2000, Boynton Beach, FL, USA) at 77K. All electrochemical measurements were performed at a CS350H electrochemical workstation (CorrTest, Wuhan, China).

## 3. Results

### 3.1. Physical Characterization of the α-Ni(OH)_2_ Nanosphere

The preparation procedure of the α-Ni(OH)_2_ nanosphere by a facile co-precipitation route is schematically depicted in Appendix A. In general, the Ni^2+^ solution was added into the urea solution drop-by-drop under unceasing agitation. Then, the mixture was transferred to a round-bottom flask with a condenser pipe, followed by heating at 105 °C, and the 3D α-Ni(OH)_2_ nanoball came into being after a while. Compared with the conventional routes for preparing spherical nanomaterials, it is worth noting that this one-step preparation process is free of both stirring and a template, which is conducive to large-scale production.

Figure 1a–d shows the surface morphology of the Ni(OH)_2_ samples performed by SEM. It is evident that the Ni(OH)_2_ nanoball is composed of interconnected and hierarchical nanoplates that have various diameters up to several micrometers. This spherical construction can provide an enlarged specific surface area for the interaction/reactions of ions, which is conducive to electron transfer, making it a desired electrode material for supercapacitors [30,31]. Notably, the globular structure changes as the reaction time increases. As seen in Figure 1a–d, the Ni(OH)_2_-24 presents an incomplete globular structure, whereas Ni(OH)_2_-36 displays an optimal morphology. However, the spherical structure gradually collapses and the nanosheets emerge as time goes on (Figure 1c,d), confirming the growth and dissolution process of the Ni(OH)_2_ nanoball. In other words, the morphology of the Ni(OH)_2_ nanosphere can be easily controlled by regulating the reaction time. In addition, TEM measurement was carried out to further confirm this unique structure. In Figure 1e, the nanostructure (2.3 μm in diameter) consists of ultrathin Ni(OH)_2_ flakes, forming a globular structure, and several lattice fringes can be observed in the HRTEM image taken from the edge of the nanosphere (Figure 1f). Moreover, EDS mapping is utilized to identify the composition of the Ni(OH)_2_ nanoball. Noticeably, the distributions of Zn and O elements, as well as C and N elements (originated from the urealysis) across the nanostructure, reveal the successful fabrication of the Ni(OH)_2_ nanoball, and the mapping result is consistent with the EDS spectrum created in Appendix A.

In order to investigate the crystal structure of the as-prepared Ni(OH)_2_ composites, the XRD patterns were recorded, and the results are presented in Appendix A. It is clear that all the samples have similar diffraction peaks, which can be indexed well to the rhombohedral phase of α-Ni(OH)_2_ (Space group: *R*, JCPDS Card No. 38-0715) with lattice constants a = 3.08 Å and c = 8.00 Å [32]. The crystal-clear peaks at 2θ = 12.40, 24.90, 33.44 and 59.36° correspond to the lattice of the (003), (006), (101) and (110) planes, respectively. The intensive peak of the (003) plane reveals the high preferred (003) orientation. Note that no other impurity peaks are observed, demonstrating the high purity of the obtained α-Ni(OH)_2_ [33]. However, the XRD reflection of the (003) and (006) planes are shifted towards high 2θ values, which may be due to the existence of the lattice distortion [34]. 

XPS measurement was utilized to analyze the chemical valence of the Ni(OH)_2_-36 compound. As shown in Figure 2a, the XPS survey spectrum indicates the presence of Ni, O and N elements, which are consistent with the EDS mapping results. The deconvolution of the Ni 2p spectrum in Figure 2b displays two major peaks with satellite peaks (labelled as Sat.) centered at 873.3 and 855.6 eV, corresponding to the Ni 2p_3/2_ and Ni 2p_1/2_ spin–orbit levels, respectively, implying the +2 valence state of the Ni ions [35]. In addition, the observed spin–energy separation between Ni 2p_3/2_ and Ni 2p_1/2_ is ~17.7 eV, which is related to the characteristics of the Ni(OH)_2_ phase [36]. The high resolution of the O 1s spectrum in Figure 2c can be divided into two components, including the nickel–oxygen bond at 531.2 eV and lattice oxygen at 528.6 eV, which is in line with previous reports [23,37]. 

The functional group information of the composites was investigated by FT-IR spectra. It is clear in Appendix A that similar peaks can be found in all Ni(OH)_2_ samples. The weak peak at 635 cm^−1^ is assigned to the Ni-O bending vibration, confirming the successful preparation of Ni(OH)_2_ [38]. The peak at ~1390 cm^−1^ is attributed to the C-O stretching vibration, indicating the intercalation of carbonate anions in the interlamination due to the dissociation of the urea [23,39]. In addition, two bands at 3445 and 1630 cm^−1^ are related to the O-H vibration from the hydroxyl groups or adsorbed water molecules of Ni(OH)_2_.

According to the above measurements, the formation process of the Ni(OH)_2_ nanosphere is proposed in Figure 3. Firstly, CO(NH_2_)_2_ releases hydroxyl ions at a high temperature, giving rise to the formation of a Ni(OH)_2_ crystal nucleus, and the corresponding reactions are suggested as follows [40,41,42]:(5)$$\mathrm{CO}{\left({\mathrm{NH}}_{2}\right)}_{2}+H_{2}O\to {\mathrm{CO}}_{2}+2{\mathrm{NH}}_{3}$$
(6)$${\mathrm{NH}}_{3}+H_{2}O\to {\mathrm{NH}}_{4}^{+}+{\mathrm{OH}}^{-}$$
(7)$${\mathrm{Ni}}^{2+}+2{\mathrm{OH}}^{-}\to \alpha \mathrm{Ni}{\left(\mathrm{OH}\right)}_{2}$$

Crystal nuclei in the solution gather to lower interfacial energy, and nanoflakes are formed due to the fast growth velocity of the (003) lattice plane. Owing to the intermolecular hydrogen bonds, the nanoflakes agglomerate and therefore generate the spherical structure to lower energy. Then, the nanosphere grows up with larger and thinner nanoplates. It is noteworthy that the nanoball will be dissolved by excess hydroxyl ions, bringing about the collapse of the spheroidal architecture (as shown in Figure 1a–d).

### 3.2. Electrochemical Performances

The electrochemical characterizations of as-prepared samples were conducted by CV, GCD, EIS and cycling performance in a three-electrode configuration. Figure 4a shows the typical CV curves at a scan rate of 10 mV/s in the potential window of 0 to 0.55 V. Unambiguously, a pair of redox peaks can be observed in all the electrodes, suggesting the characteristic faradic reaction derived from Ni(OH)_2_, and the corresponding redox reaction is as follows [43]:(8)$$\mathrm{Ni}{\left(\mathrm{OH}\right)}_{2}+{\mathrm{OH}}^{-}↔\mathrm{NiOOH}+H_{2}O+e^{-}$$

In contrast with other samples, the integral area of the CV curve for Ni(OH)_2_-36 is much higher, suggesting the prominently enhanced specific capacitance. Further, the CV curves with stable shapes of different samples at various scan rates reveal the good faradaic pseudocapacitance of the Ni(OH)_2_ samples (Figure 4b and Appendix A). Figure 4b supports the result of the CV curve, in which the Ni(OH)_2_-36 electrode has a longer discharge time than other electrodes, demonstrating a better specific capacitance. Figure 4d and Appendix A display the GCD curves of all electrodes at different current densities. It was found that a complete charging and discharging process can be implemented even at a high current density, implying the good reversibility of the redox process [44]. The voltage platforms in the charge/discharge process indicate the existence of Faradic redox reactions. The mass-specific capacitance (*C_m_*) can be calculated using the following equation [45]:(9)$$C_{m}=\frac{I\times \Delta t}{m\times \Delta V}$$
where *I* is the discharge current (A), *m* is the mass of the active materials in the electrode (g), ∆*t* is the discharge time (s), and ∆*V* is the potential window (V).

According to Equation (9), the results of the calculation for the *C_m_* are represented in Figure 4e. It is expected that the *C_m_* of Ni(OH)_2_-36 (1243 F/g) at 1 A/g is much higher compared with the Ni(OH)_2_-24 (560 F/g), Ni(OH)_2_-48 (524 F/g) and Ni(OH)_2_-60 (223 F/g) electrodes, exhibiting the excellent electrochemical performance of Ni(OH)_2_-36. Compared with single Ni(OH)_2_-based materials from previous reports (Appendix A), this specific capacitance value is competitive, even better than some Ni(OH)_2_-based composites. Further, the capacitance retention of Ni(OH)_2_-36 at 20 A/g is up to 44.07%, which is greater than Ni(OH)_2_-24 (42.86%), Ni(OH)_2_-48 (11.55%) and Ni(OH)_2_-60 (15.17%). The enhanced specific capacitance of Ni(OH)_2_-36 is probably due to the specific area resulting from the ball texture, which is verified by the N_2_ adsorption/desorption isotherms in Appendix A. The average pore sizes of all samples range from 3.401 to 3.807 nm and can be considered mesoporous materials [46]. Remarkably, the Ni(OH)_2_-36 exhibits a larger BET-specific surface area of 119.4 m^2^/g than Ni(OH)_2_-24 (42.0 m^2^/g), Ni(OH)_2_-48 (59.2 m^2^/g) and Ni(OH)_2_-60 (37.1 m^2^/g). A larger specific surface area of Ni(OH)_2_-36 can provide more active sites and motivate the diffusion of ions into inner materials, which will lead to higher capacitance [47].

Figure 4f depicts the cycling stability of Ni(OH)_2_-36 at 10 A/g after 1500 cycles with a potential of 0–0.55 V. A low capacitance retention of 40.0% is observed the in Ni(OH)_2_-36 electrode after 1500 continuous charge and discharge cycles; the reason for this is the poor conductivity of the nickel hydroxide itself. The rate performance has been investigated by GCD measurement. As shown in Figure 4g, the Ni(OH)_2_-36 electrode experiences 10 consecutive cycles at various current densities (1, 2, 3, 5, 10 and 20 A/g), then comes back to 1 A/g to run another 10 cycles. Advantageously, the specific capacitance Ni(OH)_2_-36 electrode declines with the increase in current densities and returns closed to its original level when the current density increases to 1 A/g, indicating good rate cycling behavior. Moreover, the electrochemical dynamics are revealed by EIS measurement, as illustrated in Figure 4h. In the Nyquist plots, a semicircle (in the high-frequency region) and linear part (in the low-frequency region) are observed in all curves. The equivalent circuit model is illustrated in the inset, as are the components of *Rs* (electrolyte resistance), *C* (Faradaic capacitance), *Rct* (charge transfer resistance) and *Wa* (Warburg resistance). According to the equivalent circuit, the corresponding parameters are listed in Appendix A. A smaller *Rct* value of 0.35 Ω is obtained from Ni(OH)_2_-36 compared with those of Ni(OH)_2_-24 (0.48 Ω), Ni(OH)_2_-48 (0.89 Ω) and Ni(OH)_2_-60 (0.99 Ω), proving the faster charge transfer of Ni(OH)_2_-36 due to the large surface area.

With the aim of examining the merits of Ni(OH)_2_-36 in the energy storage device, the Ni(OH)_2_-36//AC asymmetrical supercapacitor, which is composed of a positive electrode (Ni(OH)_2_-36), a negative electrode (AC), separator and electrolyte (6M KOH), is assembled in a Swagelok cell with a “sandwich” structure, and the corresponding configuration is diagramed in Figure 5a. For an asymmetrical supercapacitor device, the mass ratio of the electrodes is vital, which can be determined by the principle of charge balance according to Equations (1) and (2). Figure 5b displays the CV curves of AC, with a potential of −1.1–0 V, and Ni(OH)_2_-36, with a potential of 0–0.55 V. The specific capacitance (*C_m_*) is calculated from the CV curves by the following equation [34]:(10)$$C_{m}=\frac{\int I\left(V\right)dV}{m\times v\times \Delta V}\;$$
where *∫I*(*V*)*dV* is the enclosed areas, *m* is the mass of active materials, and ∆*V* is the potential window, respectively. Based on Equation (10), the obtained *C_m_* of AC and Ni(OH)_2_-36 at 10 mV/s are 158.62 and 437.07 F/g, respectively, and the mass ratio of AC to Ni(OH)_2_-36 is suggested to be ~2.8 in the ASC. In addition, there are no explicit polarization curves in both samples, indicating an ideal capacitive behavior with a total potential window of 1.65 V.

The CV curves ranging from 5 to 50 mV/s in Figure 5c reveal that the as-prepared Ni(OH)_2_-36//AC ASC represents common contributions of both EDLC and pseudocapacitance, and the shape of the curve is maintained evenly at 50 mV/s, implying good reversibility. Figure 5d shows the GCD curves at various current densities, and the *C_m_* are calculated as 70.1, 61.2, 57.7, 52.7, 44.8, 35.8 and 30.0 F/g at 1, 2, 3, 5, 10, 15 and 20 A/g, respectively. According to Equations (3) and (4), the Ragone plots of the as-obtained Ni(OH)_2_-36//AC ASC are given in Figure 5e, and the details can be found in Table 1. It was observed that a maximum energy density of 26.50 Wh/kg can be achieved with a power density of 0.82 kW/kg, and a maximum power density can reach up to 16.50 kW/kg, whereas the energy density still maintains 11.34 Wh/kg, confirming the outstanding rate performance of our Ni(OH)_2_-36//AC ASC. Impressively, these results outperform those of a Ni(OH)_2_-based supercapacitor, such as α-Ni(OH)_2_ nanosheet//AC (22.00 Wh/kg at 0.80 kW/kg) [24], 3D-ICHA α-Ni(OH)_2_//AC (14.86 Wh/kg at 0.14 kW/kg) [11], Co(OH)F/Ni(OH)_2_//AC (13.80 Wh/kg at 0.53 kW/kg) [42], SEP/Ni(OH)_2_//AC (24.00 Wh/kg at 0.95 kW/kg) [22] and Ni(OH)_2_/PNTs//Ni(OH)_2_/PNTs (18.80 Wh/kg at 0.41 kW/kg) [48]. Figure 5f shows the cycling performance of the ASC. It is emphasized that a slight increase in the *C_m_* before 200 cycles is probably due to the augmentation of electrochemically active sites, and a steady level is retained in the following cycles. After 4700 continual cycles at 5 A/g, this ASC device still has a remarkable capacitance retention of 85.9%, revealing the high electrochemical stability that can allow it to act as an energy storage device. Furthermore, Nyquist plots of the Ni(OH)_2_-36//AC ASC before and after cycles are created in Appendix A. The *Rct* value before the cycles decreases to 3.38 Ω, showing notable conductivity and fast ion diffusion. However, it grows to 27.08 Ω after 4700 cycles, resulting in the decline of specific capacitance.

For the purpose of investigating practical applications, two Ni(OH)_2_-36//AC ASCs were assembled in series to light up a red lamp (2.5 V), as presented in Figure 6a. Note that the small lamp gives off light even after 60 min. Furthermore, as displayed in Figure 6b, a flexible LED lamp band with a Bluetooth controller (the output rated power of 25 W) was powered by two SSCs in a series, and the word “JCUT” was shown for a few seconds, fully proving the large potential application of the energy device.

## 4. Conclusions

In summary, we have put forward a facile strategy for the preparation of a 3D α-Ni(OH)_2_ nanosphere without any template agents as a high-performance electrode material for advanced supercapacitors. By making full use of this spherical architecture, the α-Ni(OH)_2_ nanosphere represents a high specific capacitance of 1243.3 F/g at 1 A/g with an excellent rate of performance, as well as acceptable cycling stability after 1500 cycles at a large current density. We supposed that the enhanced electrochemical performance results from the unique morphology of the 3D nanosphere structure, which enables the supply of a large surface area and improves the diffusion and migration of ions in the electrochemical process. Moreover, the assembled Ni(OH)_2_-36//AC ASC shows an outstanding cycling stability of 84.7% after 4700 cycles, and it yields an impressive energy density of 26.50 Wh/kg with a power density of 0.82 kW/kg, which slightly outperforms the reported Ni(OH)_2_-based supercapacitors. Subsequently, practical applications are further investigated. Surprisingly, two ASCs in series can light a 2.5 V red lamp for more than 60 min, and an LED band with a rated power of 25 W can be powered successfully. In consideration of the facile and low-cost preparation and the remarkable electrochemical properties, the 3D α-Ni(OH)_2_ nanospheres may hold great potential applications for supercapacitors. More importantly, this facile strategy can be a reference for the fabrication of other spherical nanomaterials for next-generation energy storage materials.

## Figures and Tables

**Figure 1 nanomaterials-12-02216-f001:**
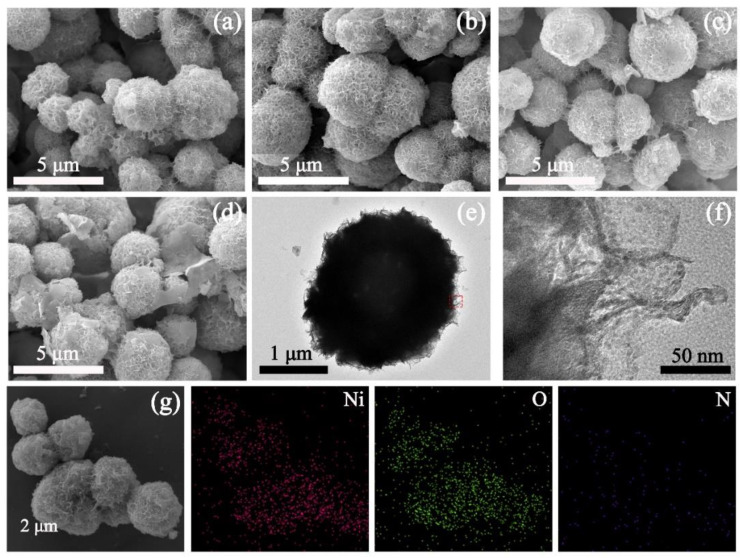
SEM images of (**a**) Ni(OH)_2_-24; (**b**) Ni(OH)_2_-36; (**c**) Ni(OH)_2_-48; (**d**) Ni(OH)_2_-60; (**e**) TEM image; (**f**) HRTEM image of Ni(OH)_2_-36; (**g**) EDS mapping of Ni(OH)_2_-36.

**Figure 2 nanomaterials-12-02216-f002:**
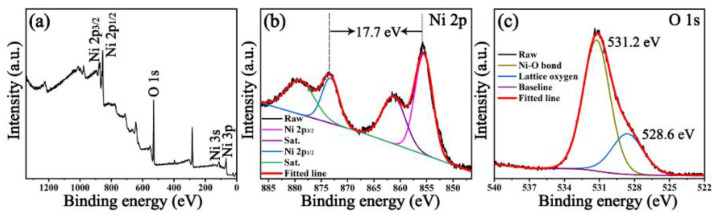
XPS spectra of (**a**) survey spectrum; high resolution of (**b**) Ni 2p and (**c**) O 1s for Ni(OH)_2_-36.

**Figure 3 nanomaterials-12-02216-f003:**
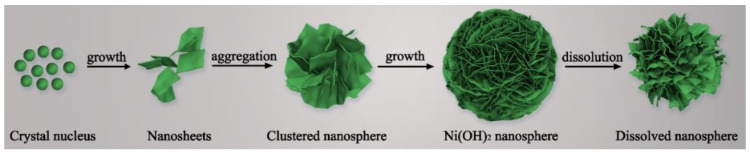
The formation mechanism of the 3D α-Ni(OH)_2_ nanosphere.

**Figure 4 nanomaterials-12-02216-f004:**
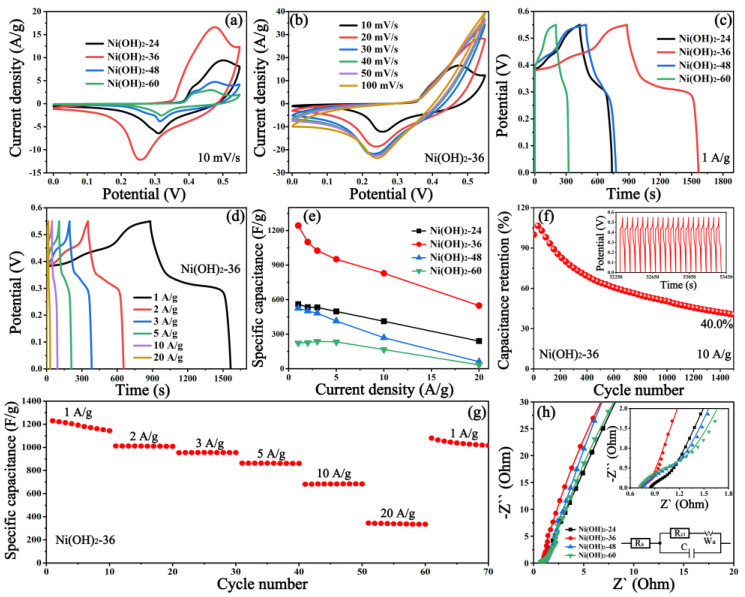
(**a**) CV curves of different samples at 10 mV/s; (**b**) CV curves of Ni(OH)_2_-36 electrode at various scan rates; (**c**) GCD curves of different samples at 1 A/g; (**d**) GCD curves of Ni(OH)_2_-36 electrode at different current densities; (**e**) obtained specific capacitance of different electrodes at various current densities; (**f**) cycling stability of Ni(OH)_2_-36 at 10 A/g after 1500 cycles, the inset shows the last 20 cycles; (**g**) rate performance of Ni(OH)_2_-36 at different current density; (**h**) Nyquist plots of the prepared electrodes, the illustrations on the upper and lower right display the enlarged Nyquist plots in the high-frequency region and equivalent circuit respectively.

**Figure 5 nanomaterials-12-02216-f005:**
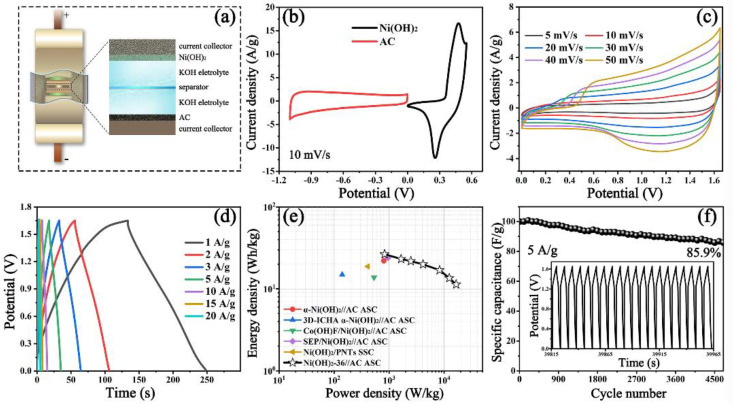
(**a**) Diagram of the as-prepared Ni(OH)_2_-36//AC ASC; (**b**) CV curves of the Ni(OH)_2_-36 and AC electrodes at 10 mV/s; (**c**) CV curves of the ASC at various scan rates; (**d**) GCD curves of the ASC at various current densities; (**e**) Ragone plots of the Ni(OH)_2_-36//AC ASC in contrast with other Ni(OH)_2_-based supercapacitor reported in previous reports; (**f**) cycling performance of the Ni(OH)_2_-36//AC ASC at 5 A/g (the inset shows the last 20 cycles).

**Figure 6 nanomaterials-12-02216-f006:**
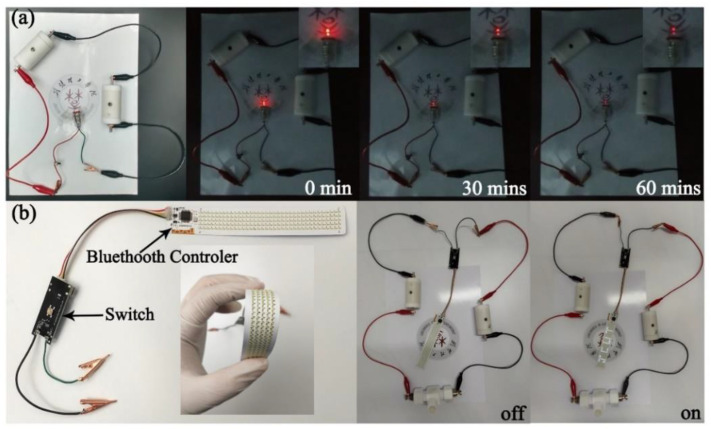
(**a**) Digital photograph of lighting the red lamp (2.5V) powered by two SSCs in series for more than 60 min; (**b**) photo of the flexible LED lamp band that was powered by three SSCs in series.

**Table 1 nanomaterials-12-02216-t001:** Performance comparison of Ni(OH)_2_-based supercapacitors from previous reports.

Supercapacitors	ΔV(V)	Conditions	C_m_(F/g)	P(W/kg)	E(Wh/kg)	Ref.
Ni(OH)_2_-36//AC	1.65	1 A/g	70	825	26.5	This work
α-Ni(OH)_2_//AC	1.60	1 A/g	248	800	22.0	[22]
3D-ICHA α-Ni(OH)_2_//AC	1.55	1.04 A/g	50.7	140	14.9	[11]
Co(OH)F/Ni(OH)_2_//AC	1.50	0.5 A/g	45	530	13.8	[42]
SEP/Ni(OH)_2_//AC	1.9	1.0 A/g	-	950	24.0	[48]
Ni(OH)_2_/PNTs	0.8	1.0 A/g	212	410	18.8	[49]

## Data Availability

Not applicable.

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
