# Peer review of "Template-Free Preparation of α-Ni(OH)2 Nanosphere as High-Performance Electrode Material for Advanced Supercapacitor"

_nanomaterials, 2022, doi:10.3390/nano12132216_

Round 1
Reviewer 1 Report
#Reviewer 1
In this work, the authors synthesize the 3D α-Ni(OH)2 nanosphere via a facile and template-free refluxing approach. The α-Ni(OH)2 nanosphere shows a specific capacitance of 1243.3 F/g at 1 A/g in 6 M KOH electrolyte and capacitive retention of 61.0% after 800 charge/discharge cycles at 10 A/g in a 3-electrode system. Furthermore, the as-assembled Ni(OH)2-36//AC asymmetric supercapacitor (ASC) shows an energy density of 26.50 Wh/kg with a power density of 0.82 kW/kg. Some of the major comments are listed below:
1. The introduction part needs to improve in the aspect of novelty and significance of this work.
2. The HR-TEM analysis related to Fig. 1 (e), and (f) should be added and correlate with the XRD analysis.
3. The amount of Ni, O, and N present in the Ni(OH)2-36 should be verified by EDS.
4. The cycle number for the stability test is very low, at least the authors should measure 5000 cycles for both the 3-electrode system and the 2-electrode system. (as presented in Fig. 4 (e), and fig. 5 (f)).
5. The authors should compare the specific capacitance of the more similar type of electrode materials for the 3-electrode system as presented in Table S2. (add 10 more references).
6. The authors should compare the specific capacitance of ASC devices with the more similar type of electrode materials and add the comparison table in the main manuscript after line 279.
7. Based on the comparison table of the ASC devices with the more similar type of electrode materials from the reported literature. the Ragone plot (Fig. (e)0, should be modified.
8. What is the correlation between the BET values and the electrochemical performance of the Ni(OH)2-36.
9. The CV of Ni(OH)2-36 should be added to Figure 4.
10. Some of the following research articles from a similar research area can be cited in the appropriate part of the introduction section:
https://doi.org/10.1016/j.jelechem.2019.113670, and https://doi.org/10.1016/j.carbon.2021.04.028
11. The authors need to show the fitted EIS plot with its inset plot and circuit diagram of the ASC device in figure 5.
12. How many Ni(OH)2-36//AC ASCs were assembled in series to light up a red lamp (2.5 V) for 60 minutes?
Reviewer 2 Report
This paper presents the fabrication and characterization of 3D α-Ni(OH)2 nanospheres prepared via a template-free dynamic refluxing approach. Products exhibit a high specific surface area of 119.4 m2 /g and abundant porous structure. The α-Ni(OH)2 electrodes as pseudocapacitive electrodes deliver a specific capacitance of 1243 F/g at 1 A/g in 6 M KOH electrolyte and acceptable capacitive retention of 61.0% after 800 charge/discharge cycles at 10 A/g. The manuscript is well organized and provide ample data, which make it worthy of publication after addressing minor corrections.
1. Overall, the English is not satisfactory. Several sentences need revision.
2. (abstract and main text) Please remove the capacitance values in decimal places. Decimal values are very difficult to reproduce.
3. (abstract) please use only one decimal for the specific surface area value.
4. (section 2.1) please provide more information about AC (particle size, porosity, tap density, etc.)
5. (section 2.2) labelling of samples is unclear. Is it temperature or duration of heat treatment at 105 °C.
6. Line 92: it seems that samples were treated for 24-60 h not 12-60 h.
7. Please quantify the loading mass of electrodes.
8. (section 2.6) please provide wavelength of X-ray sources used in XRD and XPS experiments.
9. Line 161: rewrite the sentence as “Figure 1(a-d) shows the surface morphology of the Ni(OH)2 samples performed by SEM analysis.”
10. Line 182: it is not “crystal morphology” but “crystal structure”;
11. Line 183: replace “XRD patterns were recorded” for “XRD test was performed”
12. In Fig. S3, the XRD reflection are shifted towards high 2θ values. Please explain. Please quantify lattice parameters and provide space group.
13. Fig. 5 should be placed after the call in the main text.
14. Line 276: delete “after calculation”
15. Values of the fit of the Nyquist curves (according the equivalent circuit) should be provided (Rs, Rct, CPE and Ws)
16. Line 283: it should be Swagelock cell instead of Shivelock cell
Round 2
Reviewer 1 Report
All the comments and suggestions are appended.